ecology, microbiology

resource influx, habitat opening, disturbance and invasion

**Author for correspondence:**
Luke Lear
e-mail: ll381@exeter.ac.uk

# Disentangling the mechanisms underpinning disturbance-mediated invasion

Luke Lear[1], Elze Hesse[1], Katriona Shea[2] and Angus Buckling[1]

[1]Department of Biosciences, University of Exeter, Penryn, Cornwall TR10 9FE, UK
[2]Department of Biology and Center for Infectious Disease Dynamics, 208 Mueller Laboratory, The Pennsylvania State University, University Park, PA 16802, USA

LL, 0000-0001-7726-9583; KS, 0000-0002-7607-8248

Disturbances can play a major role in biological invasions: by destroying biomass, they alter habitat and resource abundances. Previous field studies suggest that disturbance-mediated invader success is a consequence of resource influxes, but the importance of other potential covarying causes, notably the opening up of habitats, have yet to be directly tested. Using experimental populations of the bacterium *Pseudomonas fluorescens*, we determined the relative importance of disturbance-mediated habitat opening and resource influxes, plus any interaction between them, for invader success of two ecologically distinct morphotypes. Resource addition increased invasibility, while habitat opening had little impact and did not interact with resource addition. Both invaders behaved similarly, despite occupying different ecological niches in the microcosms. Treatment also affected the composition of the resident population, which further affected invader success. Our results provide experimental support for the observation that resource input is a key mechanism through which disturbance increases invasibility.

## 1. Introduction

Biological invasions are a major global issue and widely accepted as the second biggest cause of extinctions after habitat loss [1]. They reduce biodiversity, change ecosystem dynamics and cause huge financial costs [1–6]. For these reasons, it is essential to understand how some exotic species can become invasive and what makes an ecosystem vulnerable to invasion [2,7,8]. A factor frequently shown to facilitate invasions is disturbances: events that, through destruction of biomass, lead to changes in resource and habitat availability [9–12].

Disturbance can potentially alter invasion success in a number of inter-related ways. Three key examples are increased resource availability (defined as required substances such as light, soil nitrogen or water [2,12,13]), habitat opening [14] and alterations in community composition [15]. The extent to which these factors occur may vary for different disturbance types and differ in their impact on community invasibility; disentangling these differences is fundamental for understanding disturbance-mediated invader success. Disturbances can increase resource availability through associated resource inputs and resident mortality [13,16,17]; this can allow invading populations to establish by reducing competition with residents [18,19]. Consequently, increased resource input is likely to particularly benefit fast-growing, generalist invaders [8,14,20–22]. The opening up of habitats reduces any advantage established residents have over invaders due to priority effects (larger population sizes and local adaptation) [23–25]; if specific habitats are opened, it is most likely that specialists will benefit over generalist invaders [8,14,20–22]. The temporary loss of resident functional diversity associated with some disturbances [15,26]

**Figure 1.** Timeline of experimental design: the four treatments: (*a*) static + KB (added resources), (*b*) homogenized + KB (full disturbance), (*c*) static + buffer (no disturbance/control) and (*d*) homogenized + buffer (opened habitat) were carried out on day 7. All treatments were invaded post-disturbance with either a WS or SM invader and replicated six times. Treatments ended 2 days later on day 9. Homogenization lasted 30 s; 2 ml of KB or buffer was added, where appropriate.

can result in less efficient resource use and vacant habitats, again promoting invasions [19,21,22,27–30].

A large proportion of previous disturbance–invasion work concentrates on resource influxes as the cause of increased invasibility, with little consideration for habitat opening (e.g. [13,31], but see [12]). Studies that do take habitat opening into account are either observational and do not actively disturb or invade communities (e.g. [8]) or indirectly alter resource availability in habitat opening treatments through consumption reduction (e.g. [12]) [13]. Here, we experimentally investigated the relative importance of two aspects of disturbance-induced biomass destruction in determining invasion success: resource availability and habitat opening, as well as their consequences for the resident population density and diversity, in experimental populations of bacteria. In order to independently manipulate these variables, biomass destruction *per se* had to be avoided.

We used the bacterium *Pseudomonas fluorescens*, which has previously been used as a model for testing the causes and consequences of diversity [32,33] and invasion biology [28,34,35]. When introduced into a spatially structured microcosm, *P. fluorescens* diversifies into three distinct morphotypes: an air–broth interface growing wrinkly spreader (WS), a broth inhabiting smooth (SM) and the rarer bottom-dwelling fuzzy spreader [32,34,36–38]. We independently manipulated two key potential consequences of disturbance in a full factorial design: habitat opening by homogenization to open the surface niche and resource abundance through directly adding nutrients. We then determined the change in resident population composition and the success of genetically marked and visually distinguishable *P. fluorescens* lacZ marked invaders. Although originally reported to be neutral [39], the lacZ marker has been found to give a fitness advantage [40]: we additionally test this. By using two different invading morphotypes, the faster-growing SM and the more spatial niche-specific WS, it was possible to test whether different invader characteristics are predictably affected by resource input and habitat opening.

## 2. Methods

### (a) Strains

Ancestral *P. fluorescens* SBW25 was grown overnight to carrying capacity in shaken glass vials (microcosms) containing 6 ml of King's medium B (KB), at 28°C with loose lids to allow oxygen transfer. This was inoculated into static microcosms and left to diversify for 7 days, before either being plated or disturbed and invaded according to treatment group and left for a further 2 days (figure 1). Microcosms plated on day 7 ($n = 12$) were used to estimate a resident density at the time of invasion of $4.17 \times 10^9$ ($\pm 1.09 \times 10^9$ SD) colony forming units (cfu). On day 9, all treated microcosms were thoroughly homogenized and frozen at −80°C in a final concentration of 25% glycerol. Samples were plated at $10^{-5}$ and $10^{-6}$ dilutions on KB agar plates containing 100 µg l$^{-1}$ of X-gal (5-bromo-4-chloro-3-indolyl-β-D-galactopyranoside [41]). For the invader, *P. fluorescens* marked with a lacZ insert [39] was grown in static KB and left to diversify for 6 days before being plated; this strain is visually distinguishable from the wild-type in the presence of X-gal due to a colour change to blue. A single SM and a WS morph were selected, grown overnight, then both frozen in glycerol stock and plated to check morphotype purity. To stop any additional resources being added to the treatments, invaders were removed from their growth medium by centrifuging and re-diluted in M9 salt solution (3 g KH$_2$PO$_4$, 6 g Na$_2$HPO$_4$, 5 g NaCl per litre), a buffer to control for volume, before addition.

### (b) Experimental design

Treatments to manipulate resource availability contrasted the addition of the growth medium KB with that of M9 buffer. Treatments to examine habitat opening involved homogenization to open the broth surface niche, in contrast with a static control. This generates a $2 \times 2$ full factorial experimental design with four treatment groups: homogenized + M9 buffer (opened habitat only), static + KB (added resources only), homogenized + KB (full disturbance: opened habitat + resource addition) and static + buffer (no disturbance/control; figure 1). Buffer (M9) was added to the control microcosms to account for increases in broth volume in the KB addition treatments. About 2000 µl of KB or buffer was added;

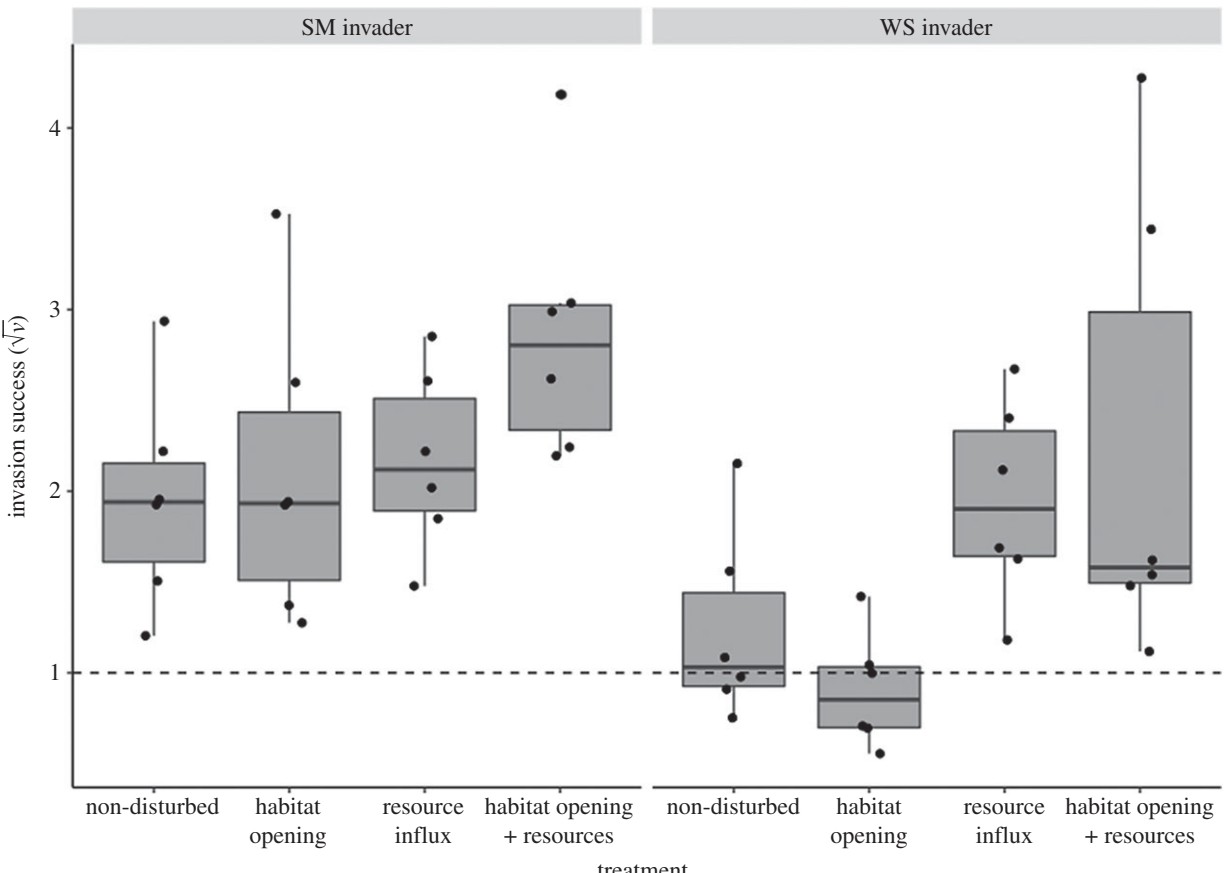

**Figure 2.** Invasion success ($\sqrt{v}$) in four treatments representing different aspects of disturbance: no disturbance, opened habitat, added resources, and combined habitat and resources. The dotted line shows equal proportional change (invader and resident fitness). The left panel shows the SM invader, right the WS. Circles represent individual microcosms.

homogenization lasted for 30 s. All microcosms were then immediately invaded with 60 µl (1% of resident population volume) of either the SM (= $1.48 \times 10^8$ cfu) or WS (= $1.21 \times 10^8$ cfu) invader. Both invader and resources were pipetted slowly down the side of the microcosm in order to minimize disruption to any biofilm. Biofilms remained whole throughout addition, with any disruption (i.e. separation from the glass) being minimal in comparison with homogenization. All treatments were replicated six times with both invaders, resulting in 48 microcosms.

## (c) Competition experiment

To determine the relative fitness of the ancestral wild-type to the ancestral lacZ strain, cultures were grown shaken overnight before 3 ml of each was mixed together in a fresh microcosm. This was plated to determine starting densities and 60 µl used to inoculate 8 KB microcosms that were then left to grow overnight statically before being plated. Relative fitness ($W$) was calculated as the ratio of growth rates, with growth estimated from Malthusian parameters (ln(end density/start density))/time [42].

## (d) Statistical analysis

Relative invader fitness (invasion success) was calculated as percentage change, $v$, of the proportion of invader (lacZ strain) relative to the resident. $v$ was given by $v = x_2(1 - x_1)/x_1(1 - x_2)$, where $x_1$ is the initial invader proportion and $x_2$ the final [43]. This was then square-root transformed. This allows comparison of proportional change even when the community is no longer growing exponentially. Generalized linear models (GLMs) were carried out in R [44] to test for the interactive effects of habitat opening, resource addition and invader morphotype on invasion success ($\sqrt{}$-transformed). We additionally added resident SM

and WS densities, plus their interaction, as covariates in this model. Similarly, we also tested for the effect of treatments and invader morphotypes on resident densities using a GLM. Resident densities were normalized using a $\log10(n + 1 \, ml^{-1})$ transformation. In both cases, non-significant model terms ($p > 0.05$) were sequentially removed and model fits were compared using $F$-tests. Relative wild-type and invader fitness were compared with a $t$-test.

## 3. Results

### (a) Invasion success

We factorially manipulated resource availability and habitat opening to determine their relative impact on the success of two different invaders. Habitat opening did not significantly affect invasion success ($F_{1,44} = 0.94$, $p = 0.34$; figure 2), nor interact with resources ($F_{1,43} = 2.04$, $p = 0.16$), but resource addition increased invasibility ($F_{1,45} = 13.0$, $p < 0.001$). Invader morphotypes differed in their success, with the mean SM invasion success ($2.78 \pm 0.73$ s.d.) higher than WS ($1.58 \pm 0.90$ s.d.) across the four treatments ($F_{1,45} = 10.8$, $p < 0.001$). There were no significant interactions between invader type and the resource and habitat manipulations in terms of invasion success ($p > 0.16$ for all interaction terms). Note that the lacZ invader increased in frequency in the majority of replicates ($v > 1$), and the lacZ ancestor had a higher relative fitness than the wild-type ancestor in competition frequencies in which starting ratios were approximately equal ($t = 5.3$, d.f. = 7, $p = 0.001$).

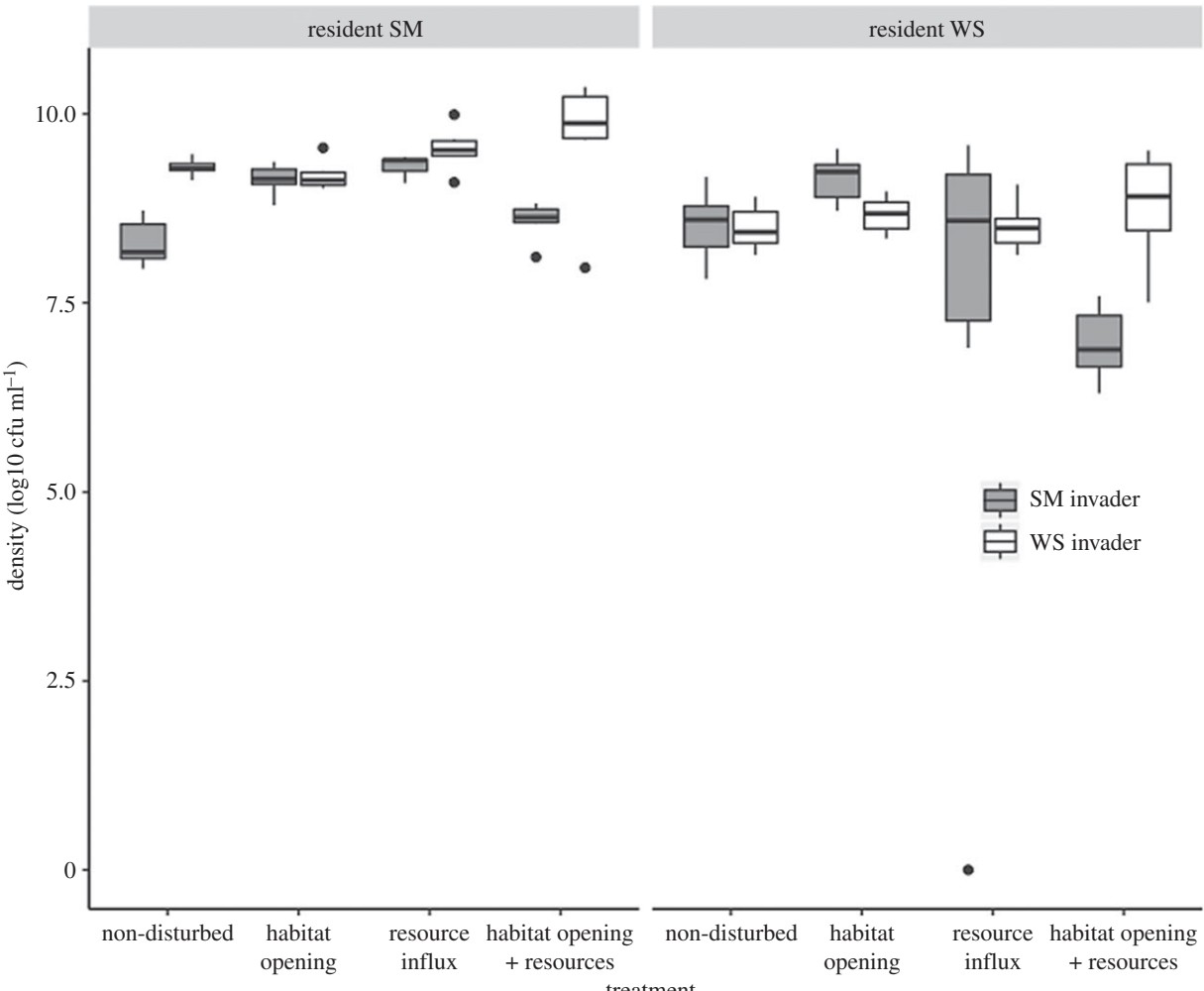

**Figure 3.** Density per millilitre of resident SM (left panel) and WS (right panel) colonies after 9 days (log10 transformed). Grey boxes show the SM invaded treatments; white the WS. Densities did not significantly differ between treatments.

## (b) Resident morph density and its effect on invasion success

Total resident density did not significantly differ between treatments ($F_{3,44} = 1.14$, $p = 0.35$; figure 3). However, specific resident morphotypes did differ. Resident SM density increased when resources were added, but only when invaded by the SM morphotype ($F_{1,44} = 7.28$, $p < 0.01$); no other variable had a significant affect. The density of resident WS was affected by a more complex three-way interaction between invader, habitat opening and resource addition ($F_{1,40} = 17.2$, $p < 0.001$).

To determine the relative importance of these changes in resident populations to the direct effects of the manipulations, we included total resident density in our model: there was a negative effect on invader success ($F_{1,43} = 17.6$, $p < 0.001$). To decompose total resident density effects into the effects of WS and SM residents, we added final resident SM and WS densities, plus the interaction between them, as covariates in the statistical model for invasion success. As before, resource addition consistently increased invasion success ($F_{1,44} = 21.7$, $p < 0.001$) and the WS invader was less successful ($F_{1,46} = 18.2$, $p < 0.001$). However, increased resident density significantly reduced invasion success (main effect of WS density and WS by SM density interaction: $F_{1,42} = 18.2$, $p < 0.001$; $F_{1,38} = 14.4$, $p < 0.001$, respectively; figure 4). Additionally, there was a significant three-way

interaction between resources, habitat opening and invader ($F_{1,37} = 8.31$, $p = 0.007$), with the greatest invasion success occurring when SM invaded microcosms that were disturbed in both ways. This demonstrates treatments had a direct effect on invasion, as well as an indirect effect through changes in resident population densities.

## 4. Discussion

Here, we experimentally determined, using a microbial system, the contributions of two distinct disturbance-induced processes predicted to enhance the success of invaders: resource influx and habitat opening. We found that resource influx provided a fitness advantage to the faster-growing invaders over the residents, while habitat opening had no impact. These results support previous observational work that attributes post-disturbance invader success primarily to resource influxes [8]. Invaders benefiting from resource influxes can be explained by reduced competition, and hence higher growth rate, allowing the invading genotypes to become established [13,18]. That the relative benefit of resource influx was independent of the life history of the invader (faster-growing SM or mat-forming WS [45]) adds to the generality of this finding.

The absence of an effect of opening up the WS ecological niche on invasion success of the WS invader may initially

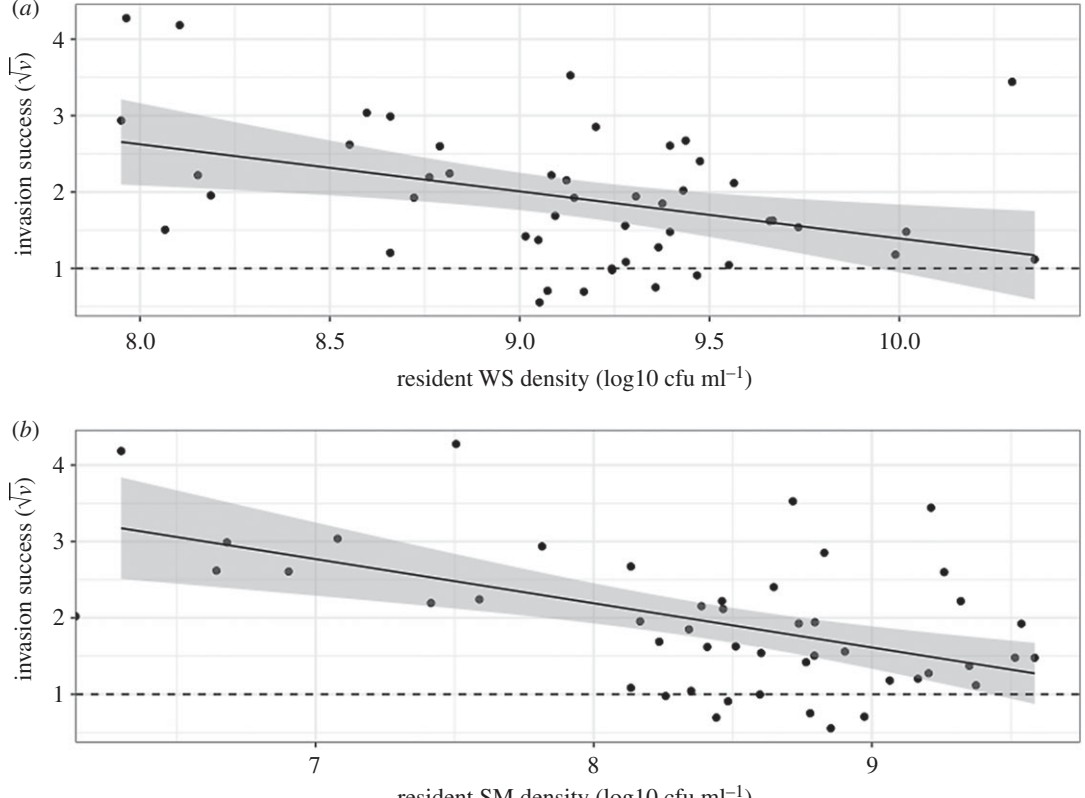

**Figure 4.** Invasion success ($\sqrt{v}$) against the log10($n$ + 1) density (cfu ml$^{-1}$) of the resident WS ($a$) and SM ($b$) morphs. Regression lines with densities as the sole explanatory variable are significant to $p < 0.02$ with the shaded area showing the 95% confidence interval. The dotted line at $v = 1$ shows equal invader and resident proportional change.

seem surprising. In this context, the resident WS were still present in the population, hence both resident and invader would have competed to colonize the opened niche, as opposed to the invader invading an already colonized niche. This lack of difference between invasion success when the residents were occupying the same niche as opposed to being present in the wider environment suggests either an absence of, or no treatment effect on, priority for this particular niche [22]. As previous work has found strong priority effects in diversified *P. fluorescens* communities, the latter is more likely [46].

The density of the resident communities also affected invader success, with the density of WS in particular having a negative effect. This is presumably because of increased resource competition. The morphotypic composition of each resident population was the result of independent evolution from an initially clonal population, and as such, there was considerable within-treatment variation. However, resident composition was also affected by interactions between the two treatments and the invader type; gaining insights into the mechanisms underlying these results may be a useful future direction. The key point is that resource influx had an indirect effect on invasion success through changes in resident community composition, in addition to the direct effect described above. Moreover, controlling for these differences revealed that SM invaders were particularly successful at invading microcosms in which resources were added and the biofilm was destroyed. More generally, the interactions between invader type and resident populations demonstrate how invaders can modify their new environment and potentially create invasion feedback loops [27,47]. Biological invasions themselves can act as disturbances; the establishment of invaders that alter resource abundances could have

a major future impact on the community's invasibility [2,18,22,29,48]. An example of this is the invasion of New Zealand forests by mammalian browsers which, through changing the forest composition, can facilitate invasion by exotic avifauna that are otherwise outcompeted by native birds [49].

While our results suggest resource influxes are a key driver of invasion success post-disturbance, the relative importance of resources and habitat opening is likely to be dependent on the community being disturbed [2,8,27]. For example, habitat opening may be expected to have a bigger effect when resident species are maladapted to the post-disturbance environment; this could be due to the disturbance itself or from niche modification by established invaders [22,24]. Maladaptation will erode both priority (fitness advantage of being the first to occupy a niche) and dominance effects (disproportionally large influence by one species on invasion resistance, usually through competitive dominance over limiting resources), ultimately reducing invasion resistance by weakening the residents' competitive dominance [27,50]. The chance of post-disturbance maladaptation occurring may be greater in more diverse communities, where species are more likely to be niche specialists [32]. Similarly, resource influxes have been shown to have a bigger effect on invasibility in communities that were resource-poor beforehand [8]. How communities respond to disturbance also depends on their previous disturbance regime [2,8,27], with deviations from this explaining twice the variation in invasion risk than disturbance *per se* [51]. Finally, invaders only being able to establish due to resource influxes raises the question of whether their populations will be stable when resource abundances return to pre-disturbance levels. For example, Petryna *et al.* [52] found that although disturbance facilitated

invasions of grassland, invader populations reduced with time after disturbance.

In summary, using a microbial system, we have provided experimental support for the proposed key role of resource influx in driving post-disturbance invasion success. However, more studies are needed to determine how disturbance history and other ecological variables will affect the generality of this conclusion.

Data accessibility. The full dataset used is accessible from the Dryad Digital Repository: https://doi.org/10.5061/dryad.j6q573n84 [53].

Authors' contributions. L.L. conceived the study, designed and carried out the experimental work, statistical analysis, interpretation of results and drafted the manuscript. E.H. analysed and interpreted the data as well as critically revised the manuscript. K.S. analysed and interpreted the data as well as critically revised the manuscript. A.B. helped with experimental design, interpretation of the data, statistical analysis and critically revised the manuscript. All authors gave final approval for publication and agree to be held accountable for the work performed therein.

Competing interests. We declare we have no competing interests
Funding. This work was supported by NSF-NERC award no. DEB-1556444.

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
