## [Reviewer comments · Proceedings of the Royal Society B: Biological Sciences]

Review History

RSPB-2019-1549.R0 (Original submission)

Review form: Reviewer 1

Recommendation

Major revision is needed (please make suggestions in comments)

Scientific importance: Is the manuscript an original and important contribution to its field?

Acceptable

General interest: Is the paper of sufficient general interest?

Good

Quality of the paper: Is the overall quality of the paper suitable?

Acceptable

Is the length of the paper justified?

Yes

Should the paper be seen by a specialist statistical reviewer?

No

Do you have any concerns about statistical analyses in this paper? If so, please specify them explicitly in your report.

No

It is a condition of publication that authors make their supporting data, code and materials available - either as supplementary material or hosted in an external repository. Please rate, if applicable, the supporting data on the following criteria.

Is it accessible?

Yes

Is it clear?

No

Is it adequate?

No

Do you have any ethical concerns with this paper?

No

Comments to the Author

1. The invasion success is here defined as the frequency of invader in the final population. A more informative measure would be the change in frequency in the population from addition to end. This would of course be problematic as it is difficult to measure the starting population size, but it would be easy to get an idea of the typical starting population size and its variation simply by plating at least 10 tubes at the same time as the invaders were added. Also of course the number of invaders should be quantified. This would give an opportunity to say something more about what happens during the invasion (for example if cell number decrease or increase). Trivially a lower resident population at the start will give a higher invasion success at the end as the starting percentage will be higher. This would need to be clarified in the manuscript.

2. I have a lot of concerns about the experimental data that would need to be explained in some way. To me it seems very odd that there can be a 200-fold difference in cfu for the same treatment (WS invader treatment 3). There is no discussion of this. In addition there is a lot of variation in the averages between very similar treatments that is not discussed. For example there is a 200-fold difference in cfu for treatment 3 depending on if the invader is a SM or WS.

3. What is the basis for the fitness increase in the lacZ strain (as shown in the competition experiment) and how large is the increase in fitness? Was part of the experiment to isolate mutants with increased fitness to use in the invasion and therefore it was allowed to diversity for 6 days before the invasion? It seems trivial that the higher fitness SM lacZ would increase more in frequency with resources added simply due to a greater number of generations of growth. As a main conclusion of the paper is to say that resource addition increase invasibility this need to be carefully dissected.

4. Even though SM lacZ is fitter and as expected can increase in density (i.e. invade) there is no reason to believe that the WS lacZ also has increased fitness relative to non-marked WS. If it does not (and it might very well have lower fitness) the expectation would in the simplest case be that it will not invade. Possibly a lower fitness genotype could invade due to a physiological advantage (i.e. not spending as long in stationary phase), but if this was the case there can also be major differences between the SM and WS invaders in this regard. It is simply not clear why there is an expectation that both the WS and SM lacZ will invade (or in fact that they even increase in frequency after being added).

5. Dryad data would ideally contain colony counts (i.e. raw data), as it seems that there were very

few for some (for example a zero for number of WS for SM invader treatment 2). It would also help to clarify that this is per 25 ul (or simply standardize to per ml or give the total population size). According to the methods 10-5 and 10-6 dilution were plated while some data say 25000, which of course would need colony counts of <1.

6. Why is no data on the amount (now it is only volume) of cells added for WS lacZ and SM lacZ included (are numbers of WS and SM similar)? This in combination with an approximate value of cfu at the time of treatments could tell us something about what happens. For example are total cell densities decreasing or increasing. Is the difference mainly in death rates or growth rates?

7. I fail to see that addition of 2 ml of liquid would not result in some disruption of the WS pellicle at the air-liquid interface. Are the pellicles not in any way attached to the walls of the tube?

8. My expectation would be that there is a strong time dependence on how successful WS types will be after habitat disruption. If the supply of nutrients are not enough to promote growth for long enough to start to form a pellicle, there will be no advantage of being WS.

Review form: Reviewer 2

Recommendation

Major revision is needed (please make suggestions in comments)

Scientific importance: Is the manuscript an original and important contribution to its field?

Good

General interest: Is the paper of sufficient general interest?

Good

Quality of the paper: Is the overall quality of the paper suitable?

Good

Is the length of the paper justified?

Yes

Should the paper be seen by a specialist statistical reviewer?

No

Do you have any concerns about statistical analyses in this paper? If so, please specify them explicitly in your report.

No

It is a condition of publication that authors make their supporting data, code and materials available - either as supplementary material or hosted in an external repository. Please rate, if applicable, the supporting data on the following criteria.

Is it accessible?

Yes

Is it clear?

Yes

Is it adequate?

Yes

Do you have any ethical concerns with this paper?

No

Comments to the Author

The authors use a model experimental system to explore the effects of disturbance on invasion success. This is an interesting experiment and I enjoyed reading the paper. I have a few comments, mainly focused many on how best to interpret the data and some unexpected results.

I'm concerned that the measure of "invasion success" (invader cfu / total cfu) used in this paper is not a good representation of "success". With this measure, an invader with the same population growth rate will appear less successful if it has invaded a community made up of more individuals (or total cfu) than if it has invaded a community with fewer individuals. This doesn't make sense to me. If every invader was inoculated into a same-sized resident community (total cfu), then this wouldn't matter, but that's not the case here. "All microcosms were then immediately invaded with 60 ul (1% of resident population volume) of either the SM or WS invader." (lines 108-110) - This means that the starting ratio of invader cfu to total cfu may have varied quite a bit. Instead of a ratio of (final invader cfu)/(final total cfu), why not just use a measure of invader population increase instead? E.g. (final invader cfu)/(initial invader cfu) . This new measure of success (invader population increase) may very well be driven by resident community size, but I think it would be better to explore this potential relationship with linear models, instead of embedding it in the measure itself.

The resident communities evolved for 7 days in the static microcosms before disturbance/ invasion. The invaders, on the other hand were isolated from a 6-day evolved community. What was the rationale for this difference? If the invader had been adapted for longer, would we expect to see a significant effect of "habitat opening" on invasion success?

The authors report that their ancestral lacZ+ strain has a higher fitness than the ancestral lacZ- strain. This is a surprising (and possibly concerning) result given that lacZ has always been described as a neutral marker in previous *Pseudomonas fluorescens* studies. Can the authors please comment on this surprising result? Is the strain used here different from that in previous studies?

Lines 171-176: If there are significant effects that depend on invader morphotype and its interactions, why not display these differences in a plot, instead of pooling everything together in Fig 3 so that we see no effects of treatment?

Lines 178-187: This paragraph is confusing because I'm not always sure which SM and WS (invader, resident, or total) are being referred to. E.g. "... we included final SM and WS densities as terms in the statistical model for invasion success." Do you mean both resident and invader? Please clarify.

Related to the statistical models discussed in this paragraph - did the authors test the effects of total cfu on invasion success? I understand that the focus here is on looking for effects of resident SM on invader SM, and resident WS on invader WS, but perhaps it's just a total population size effect? Particularly since resource influx seems to be the important treatment here and those resources are used by the entire community.

Fig 4 caption: "... with the shaded area showing the 95% confidence interval." - I don't see a shaded area on these plots.

Lines 218-220: "This lack of difference between invasion success when the residents were occupying the niche as opposed to being present in the wider environment suggests an absence of priority for this particular niche (22)." This is pretty surprising to me (as the authors

acknowledge it may be), especially given previous studies done in this system (albeit testing priority effects in different ways from this study) that do suggest there are priority effects (e.g. Zee & Fukami 2018). Can the authors comment on this finding in the context of previous work in this system?

Decision letter (RSPB-2019-1549.R0)

29-Jul-2019

Dear Mr Lear:

I am writing to inform you that your manuscript RSPB-2019-1549 entitled "Disentangling the mechanisms underpinning disturbance-mediated invasion" has, in its current form, been rejected for publication in Proceedings B.

This action has been taken on the advice of referees, who have recommended that substantial revisions are necessary. With this in mind we would be happy to consider a resubmission, provided the comments of the referees are fully addressed. However please note that this is not a provisional acceptance.

Sincerely,
Professor Gary Carvalho
<mailto:proceedingsb@royalsociety.org>

Associate Editor
Comments to Author:

Your ms has been reviewed by 2 expert reviewers. Both find the research questions worthwhile and the findings potentially interesting. However, some important technical issues with the experiments are raised that will require some additional work. The main problem identified by both reviewers is with your measure of invasion success (end-point proportion of the invader)

which instead needs to be recalculated as a growth rate (e.g. Malthusian parameter). There are a number of other issues raised by the reviewers that require additional clarification/explanation.

Reviewer(s)' Comments to Author:

Referee: 1

Comments to the Author(s)

1. The invasion success is here defined as the frequency of invader in the final population. A more informative measure would be the change in frequency in the population from addition to end. This would of course be problematic as it is difficult to measure the starting population size, but it would be easy to get an idea of the typical starting population size and its variation simply by plating at least 10 tubes at the same time as the invaders were added. Also of course the number of invaders should be quantified. This would give an opportunity to say something more about what happens during the invasion (for example if cell number decrease or increase). Trivially a lower resident population at the start will give a higher invasion success at the end as the starting percentage will be higher. This would need to be clarified in the manuscript.

2. I have a lot of concerns about the experimental data that would need to be explained in some way. To me it seems very odd that there can be a 200-fold difference in cfu for the same treatment (WS invader treatment 3). There is no discussion of this. In addition there is a lot of variation in the averages between very similar treatments that is not discussed. For example there is a 200-fold difference in cfu for treatment 3 depending on if the invader is a SM or WS.

3. What is the basis for the fitness increase in the lacZ strain (as shown in the competition experiment) and how large is the increase in fitness? Was part of the experiment to isolate mutants with increased fitness to use in the invasion and therefore it was allowed to diversity for 6 days before the invasion? It seems trivial that the higher fitness SM lacZ would increase more in frequency with resources added simply due to a greater number of generations of growth. As a main conclusion of the paper is to say that resource addition increase invasibility this need to be carefully dissected.

4. Even though SM lacZ is fitter and as expected can increase in density (i.e. invade) there is no reason to believe that the WS lacZ also has increased fitness relative to non-marked WS. If it does not (and it might very well have lower fitness) the expectation would in the simplest case be that it will not invade. Possibly a lower fitness genotype could invade due to a physiological advantage (i.e. not spending as long in stationary phase), but if this was the case there can also be major differences between the SM and WS invaders in this regard. It is simply not clear why there is an expectation that both the WS and SM lacZ will invade (or in fact that they even increase in frequency after being added).

5. Dryad data would ideally contain colony counts (i.e. raw data), as it seems that there were very few for some (for example a zero for number of WS for SM invader treatment 2). It would also help to clarify that this is per 25 ul (or simply standardize to per ml or give the total population size). According to the methods 10-5 and 10-6 dilution were plated while some data say 25000, which of course would need colony counts of <1.

6. Why is no data on the amount (now it is only volume) of cells added for WS lacZ and SM lacZ included (are numbers of WS and SM similar)? This in combination with an approximate value of cfu at the time of treatments could tell us something about what happens. For example are total cell densities decreasing or increasing. Is the difference mainly in death rates or growth rates?

7. I fail to see that addition of 2 ml of liquid would not result in some disruption of the WS pellicle at the air-liquid interface. Are the pellicles not in any way attached to the walls of the tube?

8. My expectation would be that there is a strong time dependence on how successful WS types will be after habitat disruption. If the supply of nutrients are not enough to promote growth for long enough to start to form a pellicle, there will be no advantage of being WS.

Referee: 2

Comments to the Author(s)

The authors use a model experimental system to explore the effects of disturbance on invasion success. This is an interesting experiment and I enjoyed reading the paper. I have a few comments, mainly focused many on how best to interpret the data and some unexpected results.

I'm concerned that the measure of "invasion success" (invader cfu / total cfu) used in this paper is not a good representation of "success". With this measure, an invader with the same population growth rate will appear less successful if it has invaded a community made up of more individuals (or total cfu) than if it has invaded a community with fewer individuals. This doesn't make sense to me. If every invader was inoculated into a same-sized resident community (total cfu), then this wouldn't matter, but that's not the case here. "All microcosms were then immediately invaded with 60 ul (1% of resident population volume) of either the SM or WS invader." (lines 108-110) - This means that the starting ratio of invader cfu to total cfu may have varied quite a bit. Instead of a ratio of (final invader cfu)/(final total cfu), why not just use a measure of invader population increase instead? E.g. (final invader cfu)/(initial invader cfu) . This new measure of success (invader population increase) may very well be driven by resident community size, but I think it would be better to explore this potential relationship with linear models, instead of embedding it in the measure itself.

The resident communities evolved for 7 days in the static microcosms before disturbance/ invasion. The invaders, on the other hand were isolated from a 6-day evolved community. What was the rationale for this difference? If the invader had been adapted for longer, would we expect to see a significant effect of "habitat opening" on invasion success?

The authors report that their ancestral lacZ+ strain has a higher fitness than the ancestral lacZ- strain. This is a surprising (and possibly concerning) result given that lacZ has always been described as a neutral marker in previous *Pseudomonas fluorescens* studies. Can the authors please comment on this surprising result? Is the strain used here different from that in previous studies?

Lines 171-176: If there are significant effects that depend on invader morphotype and its interactions, why not display these differences in a plot, instead of pooling everything together in Fig 3 so that we see no effects of treatment?

Lines 178-187: This paragraph is confusing because I'm not always sure which SM and WS (invader, resident, or total) are being referred to. E.g. "... we included final SM and WS densities as terms in the statistical model for invasion success." Do you mean both resident and invader? Please clarify.

Related to the statistical models discussed in this paragraph – did the authors test the effects of total cfu on invasion success? I understand that the focus here is on looking for effects of resident SM on invader SM, and resident WS on invader WS, but perhaps it's just a total population size effect? Particularly since resource influx seems to be the important treatment here and those resources are used by the entire community.

Fig 4 caption: "... with the shaded area showing the 95% confidence interval." – I don't see a shaded area on these plots.

Lines 218-220: "This lack of difference between invasion success when the residents were

occupying the niche as opposed to being present in the wider environment suggests an absence of priority for this particular niche (22).” This is pretty surprising to me (as the authors acknowledge it may be), especially given previous studies done in this system (albeit testing priority effects in different ways from this study) that do suggest there are priority effects (e.g. Zee & Fukami 2018). Can the authors comment on this finding in the context of previous work in this system?

Author's Response to Decision Letter for (RSPB-2019-1549.R0)

See Appendix A.

RSPB-2019-2415.R0

Review form: Reviewer 1

Recommendation

Accept as is

Scientific importance: Is the manuscript an original and important contribution to its field?

Excellent

General interest: Is the paper of sufficient general interest?

Excellent

Quality of the paper: Is the overall quality of the paper suitable?

Good

Is the length of the paper justified?

Yes

Should the paper be seen by a specialist statistical reviewer?

No

Do you have any concerns about statistical analyses in this paper? If so, please specify them explicitly in your report.

No

It is a condition of publication that authors make their supporting data, code and materials available - either as supplementary material or hosted in an external repository. Please rate, if applicable, the supporting data on the following criteria.

Is it accessible?

Yes

Is it clear?

Yes

Is it adequate?

Yes

Do you have any ethical concerns with this paper?

No

Comments to the Author

The authors have addressed my main concerns and questions, and improved the manuscript.

Review form: Reviewer 2

Recommendation

Accept with minor revision (please list in comments)

Scientific importance: Is the manuscript an original and important contribution to its field?

Excellent

General interest: Is the paper of sufficient general interest?

Good

Quality of the paper: Is the overall quality of the paper suitable?

Good

Is the length of the paper justified?

Yes

Should the paper be seen by a specialist statistical reviewer?

No

Do you have any concerns about statistical analyses in this paper? If so, please specify them explicitly in your report.

No

It is a condition of publication that authors make their supporting data, code and materials available - either as supplementary material or hosted in an external repository. Please rate, if applicable, the supporting data on the following criteria.

Is it accessible?

Yes

Is it clear?

Yes

Is it adequate?

Yes

Do you have any ethical concerns with this paper?

No

Comments to the Author

Overall, I am satisfied with how the authors have addressed my concerns from the previous submission. In particular, I appreciate the change made to their measure of invasion success as

well as the additional point the authors now include in their discussion of presence/ absence of priority effects in this study.

I have a couple of minor comments:

1) On the previous submission, I asked why the invaders were isolated after 6 days. I don't think the authors have answered my question. What I was trying to get at was: why were the invaders isolated after 6 days of evolution instead of 7? The invaders were then placed into microcosms that have been evolving for 7 days, so why weren't the invaders allowed to pre-evolve for 7 days as well? It just seems like a strange mis-match in the experiment. Perhaps it's the case that a single day difference in amount of pre-evolution is not enough to matter, but I'd expect that a better adapted invader would have more of an advantage in the "habitat opening" treatment.

2) I do think that an un-pooled version of Fig 3 would be more appropriate. Even if the effects are complex, I think these are more interesting and informative than the current version of the plot. I'd suggest overlaying or placing the SM invader and WS invader data side-by-side on the same panel. This means you'll present all the data on just two panels instead of four and makes it easier for comparison across treatments. If the authors still think this plot is unclear/ too complex, I'd suggest that the complete version should at minimum be included as supplementary information.

3) I still don't see a shaded area around the regression line in Fig 4. This is not a printer issue - I've only been viewing the manuscript on computer screens. Maybe it's some kind of file conversion issue, but something that should be checked.

Decision letter (RSPB-2019-2415.R0)

07-Nov-2019

Dear Mr Lear:

Your manuscript has now been peer reviewed and the reviews have been assessed by an Associate Editor. The reviewers' comments (not including confidential comments to the Editor) and the comments from the Associate Editor are included at the end of this email for your reference. As you will see, the reviewers and the Editors have raised some concerns with your manuscript and we would like to invite you to revise your manuscript to address them.

Research ethics:

Use of animals and field studies:

Please submit a copy of your revised paper within three weeks. If we do not hear from you within this time your manuscript will be rejected. If you are unable to meet this deadline please let us know as soon as possible, as we may be able to grant a short extension.

Best wishes,
 Professor Gary Carvalho
 mailto: proceedingsb@royalsociety.org

Associate Editor Board Member

Comments to Author:

Both reviewers are broadly satisfied with the revision made to the ms, although reviewer 2 raises a few important additional points. These should be addressed, in particular: (1) I agree that an unpooled version of the figure should be made available either in the main text or, at the very least, in the ESM; (2) I also am unable to see the shaded 95% CI

Reviewer(s)' Comments to Author:

Referee: 1

Comments to the Author(s).

The authors have addressed my main concerns and questions, and improved the manuscript.

Referee: 2

Comments to the Author(s).

Overall, I am satisfied with how the authors have addressed my concerns from the previous submission. In particular, I appreciate the change made to their measure of invasion success as well as the additional point the authors now include in their discussion of presence/ absence of priority effects in this study.

I have a couple of minor comments:

- 1) On the previous submission, I asked why the invaders were isolated after 6 days. I don't think the authors have answered my question. What I was trying to get at was: why were the invaders isolated after 6 days of evolution instead of 7? The invaders were then placed into microcosms that have been evolving for 7 days, so why weren't the invaders allowed to pre-evolve for 7 days as well? I just seems like a strange mis-match in the experiment. Perhaps it's the case that a single day difference in amount of pre-evolution is not enough to matter, but I'd expect that a better adapted invader would have more of an advantage in the "habitat opening" treatment.
- 2) I do think that an un-pooled version of Fig 3 would be more appropriate. Even if the effects are complex, I think these are more interesting and informative than the current version of the plot. I'd suggest overlaying or placing the SM invader and WS invader data side-by-side on the same panel. This means you'll present all the data on just two panels instead of four and makes it easier for comparison across treatments. If the authors still think this plot is unclear/ too complex, I'd suggest that the complete version should at minimum be included as supplementary information.
- 3) I still don't see a shaded area around the regression line in Fig 4. This is not a printer issue - I've only been viewing the manuscript on computer screens. Maybe it's some kind of file conversion issue, but something that should be checked.

Author's Response to Decision Letter for (RSPB-2019-2415.R0)

See Appendix B.

Decision letter (RSPB-2019-2415.R1)

07-Jan-2020

Dear Mr Lear

I am pleased to inform you that your manuscript entitled "Disentangling the mechanisms underpinning disturbance-mediated invasion" has been accepted for publication in Proceedings B.

Open Access

Your article has been estimated as being 6 pages long. Our Production Office will be able to confirm the exact length at proof stage.

Paper charges

Sincerely,

Professor Gary Carvalho

Appendix A

Response to referee's comments.

Dear Professor Carvalho,

We are very grateful to the insightful comments from the Associate Editor and reviewers, which has improved our manuscript. Please find our point by point responses below. Where both referees made similar points, their comments have been added together.

Measure of invasion success

Referee 1. The invasion success is here defined as the frequency of invader in the final population. A more informative measure would be the change in frequency in the population from addition to end. This would of course be problematic as it is difficult to measure the starting population size, but it would be easy to get an idea of the typical starting population size and its variation simply by plating at least 10 tubes at the same time as the invaders were added. Also of course the number of invaders should be quantified. This would give an opportunity to say something more about what happens during the invasion (for example if cell number decrease or increase). Trivially a lower resident population at the start will give a higher invasion success at the end as the starting percentage will be higher. This would need to be clarified in the manuscript.

Referee 2. I'm concerned that the measure of "invasion success" (invader cfu / total cfu) used in this paper is not a good representation of "success". With this measure, an invader with the same population growth rate will appear less successful if it has invaded a community made up of more individuals (or total cfu) than if it has invaded a community with fewer individuals. This doesn't make sense to me. If every invader was inoculated into a same-sized resident community (total cfu), then this wouldn't matter, but that's not the case here. "All microcosms were then immediately invaded with 60 ul (1% of resident population volume) of either the SM or WS invader." (lines 108-110) - This means that the starting ratio of invader cfu to total cfu may have varied quite a bit. Instead of a ratio of (final invader cfu)/(final total cfu), why not just use a measure of invader population increase instead? E.g. (final invader cfu)/(initial invader cfu) . This new measure of success (invader population increase) may very well be driven by resident community size, but I think it would be better to explore this potential relationship with linear models, instead of embedding it in the measure itself.

Response: Thank you for raising this point — we agree our measure of invader success could have been confounded by differences in starting ratios of the residents and invaders. We therefore have changed our measure to v — the proportional change in invader density (Ross-Gillespie, Gardner et al. 2007) between the time of inoculation and the end of the experiment. We already had directly measured starting invader density from plate counts, which was the same for all replicates for each of the invaders, as the same source populations were used. Unfortunately, as recognised by Referee 1, it was impossible to directly measure starting resident density. This would require disrupting the biofilms of the diversified microcosms at the time of invasion; and biofilm disruption was one of our experimental treatments. In order to get an idea of day-7 resident density to fulfil this measure, we followed Referee 1's suggestion of re-evolving the residents (12 replicates) for 7 days under the same conditions as the original experiment, and using this mean density for all starting resident densities. Given that all resident communities for all treatments were propagated under the same conditions prior to invasion, using this mean value will not create any systematic bias across treatments,

but instead increase the error estimate. The equation for v and reasoning why we chose this measure has been added to the manuscript (lines 176-180).

Note that we had already included resident density in linear models to explore how resident density (total density, and the density of the different evolved morphotypes) affected invasion success in the second section of the results.

Experimental data

Referee 2. I have a lot of concerns about the experimental data that would need to be explained in some way. To me it seems very odd that there can be a 200-fold difference in cfu for the same treatment (WS invader treatment 3). There is no discussion of this. In addition there is a lot of variation in the averages between very similar treatments that is not discussed. For example there is a 200-fold difference in cfu for treatment 3 depending on if the invader is a SM or WS.

Response: Within-treatment variation in resident morphotype densities is to be expected, as the morphotypic composition of the populations was the result of evolution from initially clonal populations. The extent of variation is comparable to other studies in which microcosms were disturbed in some way (e.g.(Buckling, Kassen et al. 2000, Buckling and Rainey 2002, Kassen, Llewellyn et al. 2004, Hall, Miller et al. 2012)). We emphasise this point in the discussion (lines 243-245).

LacZ fitness advantage

Referee 1. What is the basis for the fitness increase in the lacZ strain (as shown in the competition experiment) and how large is the increase in fitness? Was part of the experiment to isolate mutants with increased fitness to use in the invasion and therefore it was allowed to diversify for 6 days before the invasion? It seems trivial that the higher fitness SM lacZ would increase more in frequency with resources added simply due to a greater number of generations of growth. As a main conclusion of the paper is to say that resource addition increase invasibility this need to be carefully dissected.

Referee 2. The authors report that their ancestral lacZ⁺ strain has a higher fitness than the ancestral lacZ⁻ strain. This is a surprising (and possibly concerning) result given that lacZ has always been described as a neutral marker in previous *Pseudomonas fluorescens* studies. Can the authors please comment on this surprising result? Is the strain used here different from that in previous studies?

Response: This is indeed an interesting finding and we thank you for highlighting that it needs more discussion within the manuscript. Unfortunately, we cannot explain the increase in fitness associated with the lacZ marker. However, this finding is not unusual with Castledine *et al*, 2019 (Castledine, Buckling et al. 2019) finding the same result when comparing multiple lacZ and wildtype clones of both SM and WS morphotypes.

Referee 1

4. Even though SM lacZ is fitter and as expected can increase in density (i.e. invade) there is no reason to believe that the WS lacZ also has increased fitness relative to non-marked WS. If it does not (and it might very well have lower fitness) the expectation would in the simplest case be that it will not invade. Possibly a lower fitness genotype could invade due to a physiological advantage (i.e. not spending as long in stationary phase), but if this was the case there can also be major differences between the SM and WS invaders in this regard. It is

simply not clear why there is an expectation that both the WS and SM lacZ will invade (or in fact that they even increase in frequency after being added).

Response: Castledine *et al*, 2019 (Castledine, Buckling et al. 2019) found lacZ strains to consistently be fitter than the wildtype regardless of morphotype. That one invader may benefit more from treatments than another is why we used two ecologically different morphotypes.

5. Dryad data would ideally contain colony counts (i.e. raw data), as it seems that there were very few for some (for example a zero for number of WS for SM invader treatment 2). It would also help to clarify that this is per 25 ul (or simply standardize to per ml or give the total population size). According to the methods 10⁻⁵ and 10⁻⁶ dilution were plated while some data say 25000, which of course would need colony counts of <1.

Response: Dryad data has been updated and standardised to per ml. Dilutions 10⁻⁵ & 10⁻⁶ were averaged with these mean values used for analysis. That values can equal <1 is due to a combination of them being per 25µl and the lower dilution does not show one morphotype.

6. Why is no data on the amount (now it is only volume) of cells added for WS lacZ and SM lacZ included (are numbers of WS and SM similar)? This in combination with an approximate value of cfu at the time of treatments could tell us something about what happens. For example are total cell densities decreasing or increasing. Is the difference mainly in death rates or growth rates?

Response: We agree this is an important issue missed — the number of invader cfu added is now in the methods section. We have also used it in combination with a 7-day resident density value to give our updated measure of success, v .

7. I fail to see that addition of 2 ml of liquid would not result in some disruption of the WS pellicle at the air-liquid interface. Are the pellicles not in any way attached to the walls of the tube?

Response: This is a valid concern that is now further discussed in the manuscript (lines 112-116). By slowly pipetting down the glass disruption is minimal with no visible breaking of the biofilm. This is significantly less disruptive than the vortexing in the homogenising treatment where no visible biofilm remains.

8. My expectation would be that there is a strong time dependence on how successful WS types will be after habitat disruption. If the supply of nutrients are not enough to promote growth for long enough to start to form a pellicle, there will be no advantage of being WS.

Response: We agree there will be a strong time limit for WS success. We also agree with your second point. As WS did reform a pellicle and it was KB growth media added we are confident enough nutrients were added.

Referee: 2

Comments to the Author(s)

The authors use a model experimental system to explore the effects of disturbance on invasion success. This is an interesting experiment and I enjoyed reading the paper. I have a

few comments, mainly focused many on how best to interpret the data and some unexpected results.

Response: Thank you.

1) The resident communities evolved for 7 days in the static microcosms before disturbance/invasion. The invaders, on the other hand were isolated from a 6-day evolved community. What was the rationale for this difference? If the invader had been adapted for longer, would we expect to see a significant effect of “habitat opening” on invasion success?

Response: Thank you for this comment. The invader community was left for six days to diversify to ensure a wrinkly spreader could be isolated from a plated community at this time point. As the invaders were fitter than the resident anyway, it is unclear why the use of better-adapted invaders might reveal a habitat opening effect.

2) Lines 171-176: If there are significant effects that depend on invader morphotype and its interactions, why not display these differences in a plot, instead of pooling everything together in Fig 3 so that we see no effects of treatment?

Response: We initially felt the same when composing the manuscript, but believed the subsequent graph was less clear than the pooled version (graph below). This of course can be included if you prefer.

3) Lines 178-187: This paragraph is confusing because I’m not always sure which SM and WS (invader, resident, or total) are being referred to. E.g. “... we included final SM and WS densities as terms in the statistical model for invasion success.” Do you mean both resident and invader? Please clarify.

Related to the statistical models discussed in this paragraph – did the authors test the effects of total cfu on invasion success? I understand that the focus here is on looking for effects of resident SM on invader SM, and resident WS on invader WS, but perhaps it's just a total population size effect? Particularly since resource influx seems to be the important treatment here and those resources are used by the entire community.

Response: Thank you for highlighting this confusion — this has been clarified now. In terms of total resident density, we included this in an initial model but then decomposed into WS and SM densities. For clarity, we include it again in the second part of the Results section.

4) Fig 4 caption: "... with the shaded area showing the 95% confidence interval." – I don't see a shaded area on these plots.

Response: This is likely due to the printer used as they are there in the submitted PDF. If that is not the case this should be rectified when attaching the graphs outside of the main text.

5) Lines 218-220: "This lack of difference between invasion success when the residents were occupying the niche as opposed to being present in the wider environment suggests an absence of priority for this particular niche (22)." This is pretty surprising to me (as the authors acknowledge it may be), especially given previous studies done in this system (albeit testing priority effects in different ways from this study) that do suggest there are priority effects (e.g. Zee & Fukami 2018). Can the authors comment on this finding in the context of previous work in this system?

Response: This is indeed a surprising finding, and we agree we should discuss it more in the context of previous work. As our findings showed no effect of treatments on priority in this system we concluded there was a surprising absence. However, we have since changed this to state there could either be an absence, or presence of strong priority effects that do not differ between treatments. We go on to say in the light of previous work (i.e. (Zee and Fukami 2018)) the latter is more likely.

References

- Buckling, A., R. Kassen, G. Bell and P. B. Rainey (2000). "Disturbance and diversity in experimental microcosms." *Nature* **408**(6815): 961.
- Buckling, A. and P. B. Rainey (2002). "The role of parasites in sympatric and allopatric host diversification." *Nature* **420**(6915): 496.
- Castledine, M., A. Buckling and D. Padfield (2019). "A shared coevolutionary history does not alter the outcome of coalescence in experimental populations of *Pseudomonas fluorescens*." *Journal of evolutionary biology* **32**(1): 58-65.
- Hall, A. R., A. D. Miller, H. C. Leggett, S. H. Roxburgh, A. Buckling and K. Shea (2012). "Diversity–disturbance relationships: frequency and intensity interact." *Biology letters* **8**(5): 768-771.
- Kassen, R., M. Llewellyn and P. B. Rainey (2004). "Ecological constraints on diversification in a model adaptive radiation." *Nature* **431**(7011): 984.
- Ross-Gillespie, A., A. Gardner, S. A. West and A. S. Griffin (2007). "Frequency dependence and cooperation: theory and a test with bacteria." *The American Naturalist* **170**(3): 331-342.
- Zee, P. C. and T. Fukami (2018). "Priority effects are weakened by a short, but not long, history of sympatric evolution." *Proceedings of the Royal Society B: Biological Sciences* **285**(1871): 20171722.

Appendix B

Response to referees

Dear Professor Carvalho,

We thank you for the invitation to resubmit our manuscript and are grateful for the additional comments made by the Associate Editor and reviewers. Please find our point by point responses below.

Referee: 1

The authors have addressed my main concerns and questions, and improved the manuscript.

Referee: 2

Overall, I am satisfied with how the authors have addressed my concerns from the previous submission. In particular, I appreciate the change made to their measure of invasion success as well as the additional point the authors now include in their discussion of presence/ absence of priority effects in this study.

I have a couple of minor comments:

1) On the previous submission, I asked why the invaders were isolated after 6 days. I don't think the authors have answered my question. What I was trying to get at was: why were the invaders isolated after 6 days of evolution instead of 7? The invaders were then placed into microcosms that have been evolving for 7 days, so why weren't the invaders allowed to pre-evolve for 7 days as well? I just seems like a strange mis-match in the experiment. Perhaps it's the case that a single day difference in amount of pre-evolution is not enough to matter, but I'd expect that a better adapted invader would have more of an advantage in the "habitat opening" treatment.

Response: The additional step of growing the invader colonies on plates before growing them again overnight in broth prompted us to reduce the amount of time invaders spent evolving in broth from 7 days to 6+1, in attempt to equalise the amount of time residents and invaders had adapted to laboratory conditions prior to starting the experiment.

2) I do think that an un-pooled version of Fig 3 would be more appropriate. Even if the effects are complex, I think these are more interesting and informative than the current version of the plot. I'd suggest overlaying or placing the SM invader and WS invader data side-by-side on the same panel. This means you'll present all the data on just two panels instead of four and makes it easier for comparison across treatments. If the authors still think this plot is unclear/ too complex, I'd suggest that the complete version should at minimum be included as supplementary information.

Response: Thank you again for raising this, especially the idea of having a two-panel plot. We have updated the graph to this as suggested (see below).

3) I still don't see a shaded area around the regression line in Fig 4. This is not a printer issue - I've only been viewing the manuscript on computer screens. Maybe it's some kind of file conversion issue, but something that should be checked.

Response: Thank you again for bringing this to our attention. We will upload the figures separately to the ms and in a different file format to before.